# Fake Resume Attacks: Data Poisoning on Online Job Platforms

## ABSTRACT

While recent studies have exposed various vulnerabilities incurred from data poisoning attacks in many web services, little is known about the vulnerability on online professional job platforms (*e.g.*, LinkedIn and Indeed). In this work, first time, we demonstrate the critical vulnerabilities found in the common HR task of matching job seekers and companies on online job platforms. Capitalizing on the unrestricted format and contents of job seekers' resumes and easy creation of accounts on job platforms, we demonstrate three attack scenarios: (1) *company promotion attack* to increase the likelihood of target companies being recommended, (2) *company demotion attack* to decrease the likelihood of target companies being recommended, and (3) *user promotion attack* to increase the likelihood of certain users being matched to certain companies. To this end, we develop an end-to-end "fake resume" generation framework, titled FRANCIS, that induces systematic prediction errors via data poisoning. Our empirical evaluation on real-world datasets reveals that data poisoning attacks can markedly skew the results of matchmaking between job seekers and companies, regardless of underlying models, with vulnerability amplified in proportion to poisoning intensity. These findings suggest that the outputs of various services from job platforms can be potentially hacked by malicious users. Our codebase is available at this anonymous link.

## CCS CONCEPTS

• **Security and privacy → Web application security**; • **Information systems → Web applications**.

## KEYWORDS

fake resume, targeted attack, data poisoning, online job platforms

**ACM Reference Format:**
Anonymous Author(s). 2018. Fake Resume Attacks: Data Poisoning on Online Job Platforms. In *Proceedings of Make sure to enter the correct conference title from your rights confirmation emai (Conference acronym 'XX)*. ACM, New York, NY, USA, 9 pages. https://doi.org/XXXXXXX.XXXXXXX

## 1 INTRODUCTION

**Data poisoning attacks** in social media and web services (*e.g.*, Twitter, Reddit, and Amazon) are important problems, where malicious users attack target machine learning models and downstream tasks by injecting adversarial data to mislead the models [3, 10, 13, 38, 42]. Despite the proliferation of data poisoning attacks

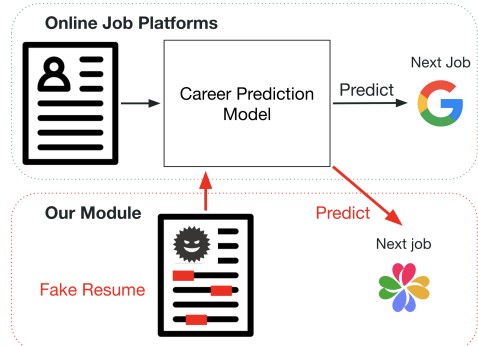

**Figure 1: An illustration of our fake resume attack.**

on online *casual* network platforms, the vulnerability of online *professional* network platforms (*e.g.*, LinkedIn and Indeed) is not well understood. As online job platforms have significantly enhanced job-seeking and hiring processes by allowing users to create their professional profiles (*i.e.*, resumes), build professional networks [7, 25], and apply these features to downstream tasks [5, 23, 27, 33], hacking popular services on such platforms would cause significant harms to both companies and job seekers alike.

In particular, one essential downstream task in the HR domain is *career prediction*, which predicts next potential job positions or companies using a user's past career trajectory. As outlined by Li et al. [16], this task provides valuable insights into potential career paths, assisting job seekers in making informed decisions about their career progression, and allowing recruiters to strategically find potential candidates who are predicted to transition into roles that align with their talent needs [18, 31, 34, 41]. This task is therefore often used for matching job seekers and companies. Conversely, however, if such a model of career prediction is manipulated, both job seekers and recruiters will be adversely affected.

On online job platforms, in general, several vulnerabilities exist: (1) it is easy to create multiple accounts of job seekers (although such clearly violates terms-of-services); (2) it is easy for job seekers to write fake experiences in their resumes (thus "fake resumes"); and (3) most of users' career trajectories that prediction models are trained with are self-reported but seldom validated due to high cost to authenticate such trajectories with official documents. A recent episode in 2022 demonstrated this vulnerability well, where 1,000 non-existent Chinese SpaceX engineers with fake profiles were found registered on LinkedIn[1]. Compared with other adversarial attacks (*e.g.*, graph adversarial attack [4, 11, 43]), therefore, a data poisoning attack via fake resumes present significant advantages to adversaries to attack (while significant challenges to online job platforms to defend), yet our understanding on the attacks and potential defenses on online job platforms is rather limited.

To mitigate this gap in understanding, using the career prediction as target downstream task, we formulate three attack scenarios: (1)

---

[1]https://www.technologyreview.com/2022/09/07/1059067/chinese-spacex-engineers-linkedin-scam/

*company promotion attack*: amplifying the likelihood of target companies in the prediction model's result; (2) *company demotion attack*: diminishing the likelihood of target companies in the prediction model's result; (3) *user promotion attack*: amplifying the likelihood of target users being matched to certain companies, and propose a novel data poisoning attack, titled **FRANCIS** (Fake Resume-based dAta poisoNing attaCks on onlIne job platformS), which generates realistic fake resumes to mislead career prediction models. Figure 1 illustrates FRANCIS. Our contributions are as follows:

- To the best of our knowledge, FRANCIS is the first to demonstrate vulnerabilities by data poisoning attacks on online job platforms.
- We formulate novel attack scenarios and a data poisoning framework to generate fake resumes focusing on the weak nature of the current online job platforms.
- Extensive experiments show that even a small fraction of poisoning can alter the prediction results regardless of underlying matchmaking and attack models.
- FRANCIS achieves improvement rates of up to 23.17 at 10% injection, 4.98 at 1%, and 1.32 at 0.1% injection.

## 2 RELATED WORK

### 2.1 Data Poisoning Attack

Attacking online platforms is often possible [2, 32], where the ultimate goal of an attacker is to exploit vulnerabilities in the platform's algorithms and generate malicious results that further their interests. [1, 12]. Data poisoning attack is one of such harmful and practical attacks [1, 38, 39], where false information and malicious inputs are injected into the dataset to train a model, resulting in biased or incorrect predictions [26]. Even though there are several works on data poisoning for web systems [19, 38, 39], attacking online professional job platforms (*e.g.*, LinkedIn and Indeed) has not been well explored. The attack on these platforms can damage both companies and users, negatively affecting both business-to-consumer and business-to-business services [9]. As a practical attack, in this work, we propose fake resume attacks on online job platforms and show the pivotal vulnerability.

### 2.2 Career Prediction

"Career prediction" is an important downstream task in the HR domain [22]. The model predicts the next potential job positions and/or companies from resumes. Liu et al. [17] used multiple social media features such as Twitter for prediction with manually defined career patterns. NEMO [16], proposed by LinkedIn, is a model to predict an employee's next career move from contextual embedding using their LinkedIn profile dataset. AHEAD [41] employs a heterogeneous company-position network to predict companies and positions simultaneously. TACTP [31] is a unified time-aware model to predict the next job with the estimated timing. NAOMI [34] is a long-term sequential model to predict the next k steps of pathways using multi-aspect embeddings and reasoning. In this work, we demonstrate the vulnerabilities of three state-of-the-art career prediction models [16, 34, 41].

### 2.3 HR-domain Downstream Tasks

There are various machine learning based downstream tasks that use resumes and career trajectory datasets in the HR domain [22].

For instance, skill extraction is a critical task for both companies and individuals, as companies want to assign their employees to the most effective department and individuals want to develop their skill sets [6, 28, 33]. Predicting employee turnover and job performance is another critical task, where models estimate the timing of employee turnover or how much they achieve based on multiple features [15, 29, 30]. Although our focus in this paper is data poisoning attack to the career prediction task, we believe that poisoned resumes could equally make other HR downstream tasks vulnerable. We leave this direction as future work.

## 3 PRELIMINARIES

### 3.1 Target Downstream Task

Online job platforms require users (*i.e.*, job seekers) to create online profiles by submitting their career histories. While these user profiles are used for various HR functions, we specifically select **career prediction**, one of the essential real-world HR tasks, as our target downstream task [16]. In this task, the model predicts an individual's subsequent job based on past job histories. Then, a job platform uses such prediction results and provide both business-to-consumer (*i.e.*, B2C) and business-to-business (*i.e.*, B2B) services: (1) For B2C side, the platform recommends a (ranked) list of companies that a job seeker matches well with, and (2) For B2B side, the platform recommends a (ranked) list of job seekers who matches well with a company so that recruiters can start recruiting actions. In other words, as the model results are used by both job seekers and companies, unique to online job platforms, it is particularly harmful if poisoned and manipulated. The details of this downstream task are explained in Section 4.5. To elucidate the repercussions of our fake resume attacks on online job platforms, we present the overview of the ecosystem in Figure 2.

### 3.2 Attack Settings

Gaining access to the specific parameters and model details of the downstream task is challenging due to their proprietary nature in commercial use. In response, we employ a black box approach by utilizing a surrogate model to generate fake resumes and then transfer it to career prediction models. For our target settings, we prefer a targeted attack approach, as it is potentially more detrimental than non-targeted attacks (*i.e.*, decreasing overall model accuracy). Further details are provided in subsequent sections. Given these settings, the attacker's knowledge base is as follows:

- The specifics of the target prediction model, including parameters and architecture, remain unknown to the attackers (black box approach).
- Attackers can only inject a limited number of fake resumes to evade the detection by the platform's security mechanism (e.g., fake resume filtering).
- It is relatively easy and cheap for attackers to create accounts on a job platform.
- For credibility, attackers usually associate their fake resumes with legitimate companies.
- All user profiles on the platform are accessible to the attackers, mirroring the visibility of professional profiles in real-world settings.

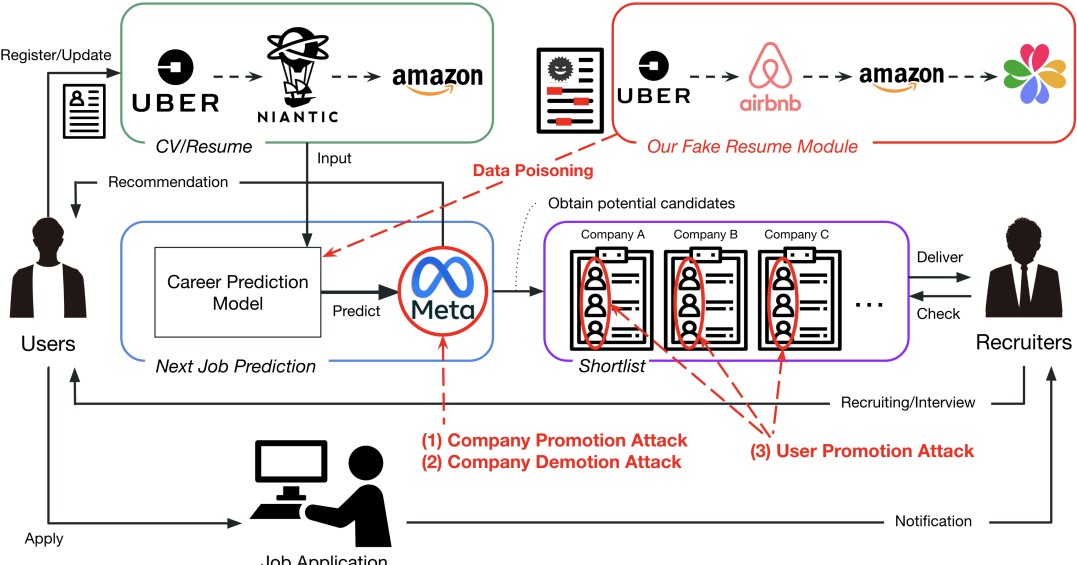

**Figure 2: Ecosystem of online job platforms and our attack scenarios. Users create their online accounts by registering their resumes, which are used for the career prediction model to predict their next career. Then, based on the predicted results, users receive the list of recommended companies as a B2C service while recruiters obtain the potential candidate lists as a B2B service. Our attack objects and scenarios are shown in red color. We propose (1) Company Promotion Attack, (2) Company Demotion Attack, and (3) User Promotion Attack. See more details in Section 3.**

## 3.3 Attack Scenarios and Objectives

Our attack objects and scenarios are highlighted in red color within Figure 2. Our fake resume attack focuses on the delivery phase of a model's prediction result. Specifically, the aim is to alter the *original* predicted companies $X$ into *target* companies $Y$, thereby influencing $Y$'s visibility to job seekers and the prominence of specific users in recruiters' shortlists. The foundation of these attacks lies in the unrestricted nature of resumes and account creation on online job platforms. We demonstrate three attack scenarios below:

**1. Company Promotion Attack**: This approach targets specific companies $X$ and artificially increases the likelihood of $X$ to be recommended to job seekers. Imagine a small company that struggles to attract talents as job seekers are often gravitated toward larger and well-known companies. Then, an attacker may offer a promotion service to such a small company, claiming that "for some \$, I can make your company to be twice more matched to job seekers than before." That is, the attacker's goal is to maximize the hit ratio of target companies. Suppose the career prediction model recommends $N$ companies to each user. We denote the fraction of users whose *top-N* recommendations include the target company after the attack. Essentially, after the attack, a significantly larger portion of users would find these target companies among their *top-N* company recommendations.

**2. Company Demotion Attack**: This approach is the inverse of the company promotion attack. Instead of increasing the likelihood, the aim is to decrease the likelihood and demote target companies. A plausible motivation is a corporate rivalry, where one company wishes to undermine the other company's presence on the platform.

**3. User Promotion Attack**: Some users, despite being keenly interested in working for specific companies, say Google or Microsoft, may lack the necessary qualifications or experience. Consequently, these job seekers are unlikely to be recommended to the recruiters of Google or Microsoft. To promote such users for specific companies, therefore, this attack seeks to manipulate model outputs, ensuring target users to be featured in the shortlists provided to target companies. Shortlist systems consist of $K$ users for each company. The goal is to maximize the averaged display rate on the shortlist, which denotes the fraction of target companies whose top-$K$ recommendations include target users.

## 3.4 Dataset

We obtained our dataset from a popular career platform[2]. From this platform, we randomly sampled resumes of job seekers who have at least five legitimate work experiences within the United States. Given that job seekers tend to pursue positions within their current position types [40], and recruiters typically seek candidates for specific roles from ones having similar experiences, we tailored our dataset selection towards two domains–the technology (Tech) and business sectors.

To construct datasets encompassing positions within these two categories, we initiated two-step pre-processing: (1) we standardized job titles in all resumes using a job title mapping model [35], which translates varying job titles into standardized ESCO-based position names [8], and (2) leveraging ESCO skill definitions [14], we filtered out positions to retain only those pertinent to technology and business sectors. To further refine our data, we filtered out companies that only appeared once in our resume dataset. After

---

[2]Details have been omitted for double-blind reviewing

## Table 1: Dataset Statistics

|  | Tech | Business |
| --- | --- | --- |
| # of resumes | 10,017 | 10,373 |
| # of unique companies | 11,679 | 12,144 |

this pre-processing, we obtained datasets with 10,017 and 10,373 resumes for tech and business sectors, respectively. Our dataset statistics are provided in Table 1. As our dataset also includes the company's general information (*e.g.*, # of employees), we label companies with less than 200 employees as "Small" and companies with more than 10,000 employees as "Large" and use them as target companies in our attacks. Figure 3 shows the statistics of companies per their sizes in our dataset.

For ethical considerations, note that any personal identifiable information (PII) in the dataset has been anonymized, retaining only career trajectories for our experiments. While we cannot publicly release our dataset due to its commercial nature, we will share our dataset for research purpose upon valid requests (*e.g.*, MOU signed).

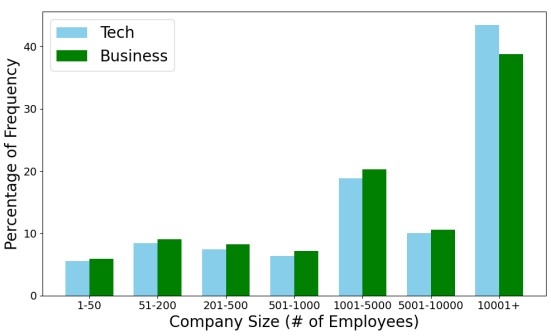

**Figure 3: Distribution of company (# of employees). We show the percentage of the sum number of companies experienced by users in the Tech and Business datasets.**

## 4 FRANCIS: OUR FAKE RESUME FRAMEWORK

In this section, we propose an end-to-end fake resume generation framework FRANCIS that induces systematic prediction errors via data poisoning. In our scenario, the attacker develops an adversarial resume generator that produces a fake resume dataset $\mathcal{D}^*$. When a model is trained with $\mathcal{D}^*$ in addition to the original data, it assists the attacker in achieving the desired behavior.

Career prediction models aim to forecast the next career a person may hold based on their professional history. Let $\mathcal{D} = \{(\mathbf{x}_i, y_i)\}_{i=1}^{N}$ be a dataset containing $N$ samples of career history data $\mathbf{x}_i$ and the corresponding next companies $y_i$, where $y_i$ belongs to a set of $M$ possible companies $\mathcal{Y}$. We denote $f : \mathbf{x} \rightarrow \mathcal{Y}$ as the career prediction model, parameterized by $\theta_f$. To attack this model, our fake resume attack approach comprises three unique modules.

### 4.1 Probabilistic Job Trajectory Generator

We design a conditional probabilistic job trajectory generator, denoted as $G$, tailored for career history data. The model, $G(\mathbf{x}_{\text{past}}, z)$, operates as a conditional sequential job trajectory generator, producing synthetic career history data, $\mathbf{x}^*$, one token at a time. The generation procedure is contingent on two primary elements:

- The career history generated up to the current point, represented as $\mathbf{x}_{\text{past}}$.
- A random latent variable, $z$.

Each token within $\mathbf{x}^*$ is derived based on a conditional probability function at every time-step $t$ until it reaches the predetermined maximum sequence length $T$. This process can be formally represented by:

$$\mathbf{x}^* = G(\mathbf{x}_{\text{past}}, z; \theta_G)$$

Here, $\theta_G$ denotes the learnable parameters intrinsic to the generator model $G$. The training objective for $G$ can be modified to incorporate this conditional generation. Consequently, our initial objective function evolves to:

$$\min_{\theta_G} \frac{1}{N} \sum_{i=1}^{N} L(f(G(\mathbf{x}_{\text{past},i}, z; \theta_G)), y_i),$$

where $\mathbf{x}_{\text{past},i}$ symbolizes the previously generated career history corresponding to the $i^{th}$ sample in the dataset.

### 4.2 Reality Regulation

We design a reality regulation function. To fabricate convincing synthetic career trajectories, our approach ensures fidelity to an underlying graph structure. Following state-of-the-art studies on formulating job transition graph [24, 34, 37, 40], we create a graph consisting the user's job transitions, in which nodes represent companies and edges are company-company transitions as shown in Figure 4. For generating a career path, each job in the sequence should be adjacent or reachable within $n$ walking steps on the graph. This adjacency constraint can be mathematically represented as:

$$\forall c_i, c_j \in \mathbf{x}^* : distance(c_i, c_j) \leq n$$

where $distance(c_i, c_j)$ computes the shortest path length between two company nodes $c_i$ and $c_j$ in the graph. Table 2 presents node degrees of large and small companies using our datasets. Average degree of large Tech companies is 42.89, of large Business companies is 36.10. Average degree of small Tech/Business companies remain at around 4 and the average degree of all Tech/Business companies keep at around 8. The average node degree in the graph varies between large and small companies.

### 4.3 Attack Module

We design an attack module to manipulate the adversarial generator $G$ to generate synthetic resumes that intentionally impact the results of the career prediction model. We follow a black box strategy rather than a white box approach as the black box strategy is more realistic (*i.e.*, the victim model is untouchable) and does not require a transparent understanding about the victim model. As such, we design our surrogate model for career prediction to produce synthetic resumes that are then utilized by the actual and unseen victim model.

Our surrogate model $f$ predicts an individual's subsequent job, aiming to optimize the following loss function:

$$L(f(\mathbf{x}_i; \theta_f)) = - \sum_{c=1}^{C} y_{ic} \log(f_c(\mathbf{x}_i; \theta_f))$$

where $C$ is the number of companies and $f_c$ is the predicted probability of company $c$.

**Table 2: Average degree of a job transition graph.**

| Company Category (# of employees) | Tech | Business |
|---|---|---|
| All companies | 9.37 | 8.49 |
| Large companies (>10k employees) | 42.89 | 36.10 |
| Small companies (<=200 employees) | 4.72 | 4.60 |

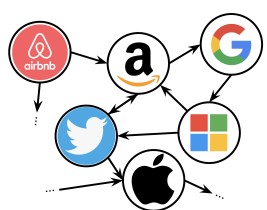

**Figure 4: An example of a job transition graph.**

For leveraging $f$ to guide $G$, we use backpropagation signals from $f$. The aim is for $G$ to generate a new resume, $x^*$, such that $f(x_i)$ (with a perturbation $y^*$) results in a targeted prediction label $L^*$ from the set of companies for $x_i$. Our optimization objective for this function is:

$$\min_{\theta_G} L^*(f(G(\mathbf{x}_i; \theta_G)))$$

### 4.4 Objective Function

We define objective functions in accordance with our three distinct attack scenarios. The attacker's goal is to craft realistic fake resumes that target the surrogate model by optimizing the objective function pertinent to each scenario.

(1) Company Promotion Attack

$$L_{\text{promotion}} = -\frac{1}{N} \sum_{i=1}^{N} \sum_{j \in T} P_{ij} \tag{1}$$

In this scenario, the objective is to maximize the likelihood of target companies being predicted.

(2) Company Demotion Attack

$$L_{\text{demotion}} = \frac{1}{N} \sum_{i=1}^{N} \sum_{j \in T} P_{ij} \tag{2}$$

The goal here is to diminish the likelihood of target companies in the surrogate model's predictions.

(3) User Promotion Attack

$$L_{\text{user-company}} = -\frac{1}{U} \sum_{i \in U} \sum_{j \in T} P_{ij} \tag{3}$$

In this attack, the aim is to enhance the likelihood of specific users (or resumes) being associated with target companies, optimizing over a select group of users, denoted as $\mathcal{U}$.

### 4.5 Surrogate Model

For the career prediction task, we adopt an RNN model with several state-of-the-art models. Following [16], we employ LSTM architecture to capture intricate patterns in job transitions that could indicate a user's future career shift.

Consider the dataset $\mathcal{D} = \{(x_i, y_i)\}_{i=1}^{N}$, where each $x_i$ is a sequence of companies, and $y_i$ is the next potential company. We have a function $\mathcal{F}(x)$ representing a multiclass classification model. Trained with the categorical cross-entropy loss, it provides a prediction probability $\hat{y}$ as:

$$\hat{y} = \mathcal{F}(x; \theta_f)$$

where $\theta_f$ is the surrogate model's parameters. Following the prediction, the top-$k$ predicted companies are:

$$\mathbf{y}_{\text{top-k}} = \text{TopK}(\hat{y})$$

Our LSTM configuration consists of two layers with 128 units in the first and 64 in the second, a dropout layer (rate of 0.5), and the Adam optimizer for loss function optimization.

## 5 EVALUATION

In this section, we discuss the evaluation results of FRANCIS and the baseline models using a real-world dataset, as detailed in Section 3.4. Our evaluation seeks to address the following Research Questions (RQ):

(1) **RQ1:** Is it feasible to poison career prediction models?
(2) **RQ2:** How does our fake resume attack perform against baseline approaches?
(3) **RQ3:** To what extent does injecting fake resumes affect the performance of career prediction?

### 5.1 Evaluation Protocol

*5.1.1 Attack Performance.* To address RQ1 and RQ2, we evaluate the efficacy of our attack to various target models, as follows.

**Degree of Attack Success:** In the context of data poisoning attacks in career prediction for online job platforms, it is important to measure how well the attacks promote or demote the target in order to measure the success rate of the attack. For this, we use the *Improvement Rate* (IR) of the average target Hit Ratio (*i.e.*, HR) in the original surrogate model as our measure. The improvement rate IR is defined as the increase in *HR* after data injection over the *HR* before data injection, as follows:

$$\text{IR@}k = \frac{\text{HR@k}_{\text{after}}}{\text{HR@k}_{\text{before}}}$$

This gives us an indication of how much we are able to manipulate the visibility of the target through data poisoning. We vary the injection ratio in our experiments to discern its impact on the attack's success. Following previous studies [36], we set $k$=10.

**Target Company and User Selection:** In the company promotion and demotion attacks, we randomly sample 100 companies from "Small", "Large", and random companies on our dataset (see Section 3.4 for the company definition), and see the average $IR@10$ for the target companies. In the user promotion attack, we set "Large" companies as target companies assuming that some users want to get an interview or any recruitment opportunity for top companies competing with other job seekers, and extract users from those who never experienced "Large" companies (we name these as "Specific" users) or sample 20% users from all users as the target users (we name this as "Random" users). Afterward, we see the average HR@10 for the target companies in the target users.

**Target Victim Models:** To attack the career prediction models, we set the three state-of-the-art models as target victim models:

NEMO [16], AHEAD [41], and NAOMI [34]. All three models are designed and experienced in the data from online job platforms or real-world resumes.

**Baseline Attack Models:** As aforementioned, to the best of our knowledge, no existing work addresses our presented task (*i.e.*, data poisoning attacks on career prediction). The most relevant work to ours is [36], however, their data poisoning attacks use alternating sequences (*i.e.*, [target, non-target, target, non-target, ...]), resulting in clearly unrealistic resumes that can be easily detected by a simple rule-based system. Consequently, we compare FRANCIS with existing methods that are most compatible.

- **Random:** This attack randomly generates job trajectories and inserts 1) one target company for the promotion attack and 2) a non-targeted company for the demotion attack.
- **Popular:** We prepare the top 10% frequent companies. Then, this model randomly generates job trajectories from those frequent companies and follows the same process of the random attack.
- **GPT-4:** GPT-4 is the latest large language model. Due to the model's robustness and generalizability in various domains [21], we assume GPT-4 may be also useful for the HR domain. We use the zero-shot approach to obtain job trajectories. Based on the impersonation strategy, we use the following prompt to make GPT-4 generate fake trajectories.

> **Prompt**: You are a professional career advisor. I'm seeking your assistance to generate realistic career trajectories for professionals in the {{tech or business}} field. Can you provide {{n}} career paths, each containing at least five job experiences? Please ensure that all company names mentioned are real-world entities. Our primary objective is to {{increase or reduce}} the likelihood of the following target companies by adding them to HR models. Target Companies List: {{target_company_list}}

Due to the output length limitation of GPT-4, we only show the injection ratio 0.1% and 1% for this baseline.

- **DQN:** Deep Q-Network (DQN) underlies an RNN architecture tailored for sequential career trajectories. This model is trained with rewards derived from the prominence and rank of target jobs within top-k predictions [20, 38]. This model is used only for the promotion attacks due to the limited nature of the loss function in the original model.

### 5.1.2 Effect of Fake Resume Injection on Downstream Task Performance.
Another challenge in injecting fake resumes is to make them indistinguishable from real resumes. If the overall career prediction after data poisoning changed much, the system would easily notice and alert it. Thus, to answer **RQ3**, we examine the performance shift in the career prediction before and after the data poisoning to see how much it affects the performance compared to the baseline attack models.

## 5.2 Result

### 5.2.1 RQ1: Attack Feasibility.
Tables 3, 4, and 5 show the results of company promotion attack, company demotion attack, and user promotion attack, respectively. In these tables, we use NEMO [16],

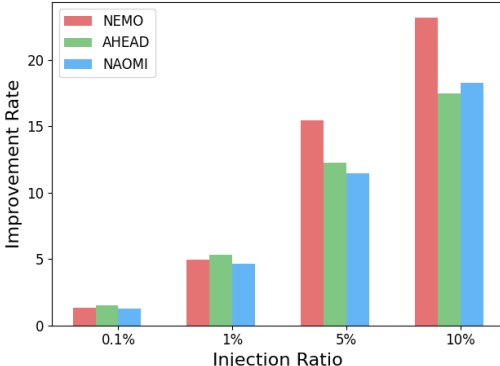

**Figure 5: Victim model's improvement rate comparison in the Tech dataset with our attack method, targeting "Small-size" company.**

LinkedIn's career prediction model, as the target victim model, and we set three steps in our reality regulation module.

**Overall:** Our results from Tables 3-5 clearly illustrate that, regardless of the datasets (*i.e.*, tech and business), attack scenarios, target companies, injection rates, and attack methods, there are more or less vulnerabilities by data poisoning on the career prediction, with the vulnerabilities amplifying in proportion to poisoning intensity. Remarkably, even minimal injections, as low as 0.1% or 1%, can induce a significant drop in the model's expected behavior.

Figure 5 shows the comparison of the improvement rate on each victim career prediction model when attacked by FRANCIS. Here, we used the Tech dataset targeting "Small" companies. We can observe that each model can be attacked successfully, and the vulnerability increases by injection ratio. In the subsequent sections, we delve deeper into the discussions of each specific attack setting on NEMO [16].

### 5.2.2 RQ2: Attack Performance Comparison.

**Company Promotion Attack:** In this attack, a higher score indicates a better outcome, which means the attack model improves the target companies' visibility. The results reveal a pronounced impact when targeting "Small" companies with 10% injection, and FRANCIS achieved an improvement rate of 23.17 and 10.48 for the corresponding Tech and Business datasets, which 2.9 times higher than best baseline in Tech domain, and 2.1 times higher than best baseline in Business domain. Even with a tiny injection amount of 0.1%, FRANCIS performance relatively improved 8.2% compared to best baseline *DQN* in Tech domain, and enhanced 13.9% compared to the best baseline *GPT-4* in Business domain.

When targeting "Large" companies, FRANCIS still outperformed all the compared baselines although the impact is diminished. This reduced vulnerability can be attributed to the prevalent representation of large companies within the data, as shown in Figure 3. While inducing a drastic enhancement remains challenging, an attack is still attainable. For instance, with only 0.1% injection, our model relatively improved 12.1% over best baseline GPT-4 in Tech domain, and 3% over the best baselines GPT-4 and DQN in Business domain. The improvement is much higher with a larger injection rate. Specifically, at 5% injection rate, our FRANCIS performance is

**Table 3: *Company Promotion Attack* - Improvement Rate IR@10 of FRANCIS vs baselines. In this table, the victim model is NEMO [16], and the adjacent step in our reality module is three. In the promotion attack, a higher score is better.**

| Target Company | Injection | Dataset | | | | | | | | | |
| --- | --- | --- | --- | --- | --- | --- | --- | --- | --- | --- | --- |
| | | Tech | | | | | Business | | | | |
| | | Random | Popular | GPT-4 | DQN | FRANCIS | Random | Popular | GPT-4 | DQN | FRANCIS |
| Small-Size | 0.1% | 0.83 | 1.10 | 0.73 | 1.22 | **1.32** | 1.00 | 1.08 | 1.08 | 1.06 | **1.23** |
| | 1% | 1.59 | 1.34 | 0.73 | 1.46 | **4.98** | 1.90 | 1.16 | 0.83 | 1.16 | **3.56** |
| | 5% | 4.49 | 4.24 | - | 1.56 | **15.46** | 2.56 | 3.49 | - | 1.20 | **9.57** |
| | 10% | 7.90 | 6.20 | - | 1.56 | **23.17** | 4.56 | 4.99 | - | 1.23 | **10.48** |
| Large-Size | 0.1% | 1.14 | 0.98 | 1.09 | 1.09 | **1.26** | 1.07 | 1.06 | 1.07 | 1.07 | **1.10** |
| | 1% | 1.33 | 1.14 | 1.53 | 1.41 | **1.67** | 1.10 | 1.06 | 1.09 | 1.15 | **1.39** |
| | 5% | 1.54 | 1.45 | - | 1.36 | **3.80** | 1.39 | 1.36 | - | 1.16 | **2.81** |
| | 10% | 1.86 | 2.04 | - | 1.41 | **3.36** | 1.63 | 1.65 | - | 1.28 | **2.88** |
| Random-Size | 0.1% | 1.00 | 1.06 | 1.09 | 1.04 | **1.13** | 1.04 | 0.91 | 0.86 | 1.04 | **1.16** |
| | 1% | 1.40 | 1.36 | 2.27 | 1.13 | **2.49** | 1.41 | 1.23 | 1.60 | 1.36 | **2.27** |
| | 5% | 2.40 | 1.95 | - | 1.31 | **6.36** | 2.89 | 2.35 | - | 1.38 | **7.90** |
| | 10% | 3.00 | 3.40 | - | 1.31 | **9.45** | 4.37 | 4.44 | - | 1.41 | **8.27** |

**Table 4: *Company Demotion Attack* - Improvement Rate IR@10 of FRANCIS vs baselines. In this table, the victim model is NEMO [16], and the adjacent step in our reality module is three. In the demotion attack, a lower score is better.**

| Target Company | Injection | Dataset | | | | | | | | | |
| --- | --- | --- | --- | --- | --- | --- | --- | --- | --- | --- | --- |
| | | Tech | | | | | Business | | | | |
| | | Random | Popular | GPT-4 | DQN | FRANCIS | Random | Popular | GPT-4 | DQN | FRANCIS |
| Small-Size | 0.1% | 1.07 | 0.83 | 1.00 | N/A | **0.73** | 1.00 | 0.90 | 1.16 | N/A | **0.73** |
| | 1% | 0.83 | 1.07 | 1.00 | N/A | **0.61** | 0.90 | 1.00 | 0.83 | N/A | **0.75** |
| | 5% | 0.98 | 0.83 | - | N/A | **0.73** | 0.73 | 0.83 | - | N/A | **0.75** |
| | 10% | 0.83 | 0.83 | - | N/A | **0.61** | 1.06 | 0.83 | - | N/A | **0.75** |
| Large-Size | 0.1% | 1.14 | 1.19 | **0.91** | N/A | 1.02 | **0.98** | 1.18 | 1.03 | N/A | 0.99 |
| | 1% | 1.08 | 1.04 | 1.06 | N/A | **0.98** | 1.07 | 1.01 | 1.09 | N/A | **0.95** |
| | 5% | 1.08 | 1.01 | - | N/A | **0.98** | 0.95 | 1.10 | - | N/A | **0.94** |
| | 10% | 1.01 | 0.98 | - | N/A | **0.93** | 1.01 | 0.98 | - | N/A | **0.94** |
| Random-Size | 0.1% | 0.95 | 1.13 | 1.09 | N/A | **0.86** | 1.28 | 0.91 | 0.86 | N/A | **0.86** |
| | 1% | 0.95 | 1.04 | 1.09 | N/A | **0.86** | 0.91 | 0.91 | 1.48 | N/A | **0.80** |
| | 5% | 0.95 | 0.85 | - | N/A | **0.82** | **0.79** | **0.79** | - | N/A | 0.80 |
| | 10% | **0.85** | 0.95 | - | N/A | 0.86 | 0.79 | 0.86 | - | N/A | **0.68** |

**Table 5: *User Promotion Attack* - Improvement Rate IR@10 of FRANCIS vs baselines. In this table, the victim model is NEMO [16], the adjacent step in our reality module is three, and the target company is "Large". As to the target users, "Specific" users are users who never experienced "Large" companies, while "Random" users are those randomly sampled 20% of all users. In the promotion attack, a higher score is better.**

| Target Users | Injection | Dataset | | | | | | | | | |
| --- | --- | --- | --- | --- | --- | --- | --- | --- | --- | --- | --- |
| | | Tech | | | | | Business | | | | |
| | | Random | Popular | GPT-4 | DQN | FRANCIS | Random | Popular | GPT-4 | DQN | FRANCIS |
| Specific Users | 0.1% | 1.00 | 1.00 | 1.03 | **1.12** | 1.11 | **1.13** | 0.97 | 0.97 | 1.06 | 0.98 |
| | 1% | 1.10 | 1.10 | 1.45 | 1.22 | **1.51** | 1.16 | 1.06 | 1.03 | 0.98 | **1.56** |
| | 5% | 1.37 | 1.63 | - | 1.21 | **2.90** | 1.47 | 1.47 | - | 1.19 | **2.45** |
| | 10% | 1.93 | 1.97 | - | 1.30 | **3.80** | 1.72 | 1.63 | - | 1.16 | **2.48** |
| Random Users | 0.1% | **1.24** | 1.20 | 1.16 | 1.08 | 1.12 | 1.08 | 0.89 | **1.22** | 1.11 | 1.17 |
| | 1% | 1.20 | 1.08 | 1.32 | 1.08 | **2.24** | 1.03 | 1.11 | 1.09 | 1.41 | **1.54** |
| | 5% | 1.52 | 1.56 | - | 1.28 | **6.64** | 1.54 | 1.32 | - | 1.39 | **2.32** |
| | 10% | 2.40 | 1.88 | - | 1.20 | **13.08** | 1.70 | 1.43 | - | 1.41 | **2.70** |

2.8 times higher than the best baseline in Tech domain, and 2 times higher than the best baseline in Business domain.

For "Random" companies, we still observe the similar improvement pattern of our proposed model. At 1% injection, our model relatively improved 9.7% over best baseline GPT-4 in Tech domain, and 41.9% over the best baselines DQN in Business domain.

Interestingly, while the GPT-4-induced synthetic resumes demonstrate some efficacy against Large-Size and Random-Size companies, they become counterproductive when targeting Small-Size companies. A plausible explanation for this phenomenon might be GPT-4's extensive training on job descriptions or corpora from renowned companies. Consequently, it could be under-equipped to generate convincing content for lesser-known or smaller companies.

**Company Demotion Attack:** In this attack, a lower score indicates a better outcome, which means the attack model reduces the target companies' visibility. Compared with the company promotion attack, the effects stemming from the company demotion attack are weak, but it still remains effective in manipulating prediction results. Also, we can observe that the Random attack is reasonably influential in this attack setting.

It's particularly evident that when "Small" companies are the target, the attack succeeds in considerably reducing the hit ratio. In contrast, attacking "Large" companies yields limited returns, with a 10% data poisoning only resulting in a modest improvement rate of around 0.93 or 0.94. This resilience can be attributed to the preponderance of large companies in the dataset, rendering the model robust against attempts to degrade prediction outcomes. This observation is consistent with the earlier finding from the company promotion attack where significant improvements are elusive for "Large" companies. When targeting Random-Size companies, the resulting impact occupies the middle.

**User Promotion Attack:** In this attack, a higher score indicates better. We set the target companies as "Large" ones, and the target users as "Specific" and "Random" users (see the detail in Section 5.1). Promoting users via fake resume attacks is also feasible. However, minor poisoning rates, such as 0.1%, yield minimal observable changes, while an injection of 1% or more can notably enhance the $HR@10$ by over 1.5 times.

It's important to note that "Specific" users are characterized by their lack of experience with "Large" companies. In contrast to "Random" users, the improvement rate for these "Specific" users is diminished. This trend can be tied back to our earlier discussions on the inherent robustness of "Large" companies. Conversely, examining the Tech data for "Random" users reveals a significant boost in the hit ratio after data poisoning. This suggests that predictions related to affiliations with giant tech companies might be heavily influenced by prior experiences with "Large" companies, implying FRANCIS may amplify users with experience with other "Large" companies to be recommended to more specific large companies.

*5.2.3* ***RQ3: Effect of Fake Resume Injection***. This section evaluates the effect of injecting fake resumes. Table 6 shows the overall performance change rate of career prediction before and after fake resume attacks. To delve deeper into the implications of fake resume injections, we conducted a series of experiments on our pre-trained career prediction model (*i.e.*, surrogate model). Specifically, we injected fake resumes generated for company promotion attack with

**Table 6: Relative change in career prediction accuracy after fake resume injection for the company promotion attack with a 1% injection ratio.**

| Attack | Tech | | | Business | | |
|---|---|---|---|---|---|---|
| | Small | Large | Rand | Small | Large | Rand |
| Random | +1.33% | +1.47% | +1.61% | +1.15% | +1.08% | +0.95% |
| Popular | +1.12% | +1.82% | +1.75% | +1.01% | +1.08% | +0.68% |
| GPT-4 | +1.33% | +1.05% | +1.26% | +0.95% | +1.01% | +0.95% |
| DQN | +1.33% | +1.47% | +1.05% | +1.08% | +0.81% | +0.95% |
| FRANCIS | +1.12% | +0.98% | +1.05% | +1.28% | +1.15% | +0.88% |
| None | +1.19% | +1.19% | +1.19% | +1.01% | +1.01% | +1.01% |

a 1% injection ratio into the model and proceeded with an additional training of 20 epochs. The primary objective was to gauge the relative improvement in performance from the original metrics post-injection. For a holistic understanding, we also implemented a comparative baseline where no additional data was introduced but the model underwent the same additional training epochs. This scenario is denoted as "None" in the Table 6.

While the Random and Popular attacks achieved improvements in the hit ratio during the company promotion attack, they exemplified a significant change rate in the career prediction model's accuracy, often deviating substantially from the standard performance. The performance shifts induced by GPT-4 and DQN were not consistent and varied based on the targeted companies. On the other hand, FRANCIS exhibited behaviors closely aligned with the original dataset. Notably, its improvement rate was contained within one standard deviation from the original (*i.e.*, "None") improvement rate. This consistency in FRANCIS's performance indicates the effectiveness of our reality regulation module, suggesting that it generates resumes that are not just synthetic but also highly realistic, closely mimicking genuine career trajectories.

## 6 LIMITATIONS AND FUTURE WORK

This study focused primarily on career positions within the realms of tech and business. It is also crucial to extend our exploration into other domains and assess performance on datasets that encompass a mix of multiple or cross-domain genres. Nonetheless, our research successfully underscores the vulnerabilities introduced by data poisoning in online job platforms. While the focus of the current investigation was career prediction, it raises concerns about potential susceptibilities in other HR downstream tasks. In the future, it would be intriguing to scrutinize how these vulnerabilities manifest across a broader spectrum of HR applications and tasks.

## 7 CONCLUSION

In this paper, we highlighted vulnerabilities in career prediction through fake resume attacks. By exploiting the flexible format of resumes and the nature of online job platforms, we presented three potential attacks: (1) company promotion attack, (2) company demotion attack, and (3) user promotion attack. We proposed a fake resume generation system that manipulates predictions through data poisoning, and showed the performance in the real-world resume datasets. This exposes the risk of online job platforms being compromised by ill-intentioned users.

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
