# OpenReview forum: "Fake Resume Attacks: Data Poisoning on Online Job Platforms"
_ACM.org/TheWebConf/2024/Conference — TheWebConf24_

### Official Review · Reviewer_3RC6 · 2023-11-18

**Novelty:** 5
**Technical Quality:** 6

**Review:**

I thank the reviewers for their submission, which is well written and clear. In this paper, the authors propose FRANCIS, a fake resume a "fake resume" generation framework that can be used by job applicants to improve their visibility on an online platform and companies to increase their visibility or decrease other companies visibility.

My main issue with this paper is that, as currently constructed, is hard to envision the real world applications of FRANCIS and the attacks shown in this publication. The authors motivate this paper based on the use of "Career predictions" algorithms. However, it is unclear whether this are widely adopted by companies (or online platforms) as part of their hiring process / recommendation systems. This needs to be better motivated, and supported by citations backing the real world adoption of these. Later on the paper, the authors use NEMO for validation, but this needs to be introduced earlier to motivate the paper.

It is also unclear how likely these approaches are to be used. An user might have an incentive to use a fake resume to increase their visibility on one of these platforms, however if chosen, they would still need to go through an interview process and in some cases even background checks to ensure that their resume and qualifications are accurate.

The authors compare their system mostly with naive approaches such as randomly generating a career path. It is unclear how much of an advantage FRANCIS would be in a real world scenario in which an user might tailor their CV based on the company they are actually trying to target.

Further more, since the authors do not know

**Questions:**

Why does naming the authors source break double blind? If they got access to an external dataset, naming this company should not break anonymity.

I don't think there's an ethical issue with this paper, but I would have liked a further discussion about the risks of using this dataset. While they anonymity PII, it looks like it would be relatively achievable to actively de-anonymize users in the dataset, Does this dataset contain any non public data (i.e., if someone were to crawl this platform would they have access to the same data as available in this dataset).

**Reviewer Confidence:**

3: The reviewer is confident but not certain that the evaluation is correct

**Scope:**

3: The work is somewhat relevant to the Web and to the track, and is of narrow interest to a sub-community

---

### Official Review · Reviewer_uynF · 2023-11-22

**Novelty:** 4
**Technical Quality:** 4

**Review:**

The paper describes three potential attacks on online job platforms that can be abused by uploading fake resumes.

Overall, the paper addresses a practical problem, and shows its feasibility on a sample real-world data set. The evaluation is presented well and is sufficiently detailed.

My main concern is how realistic the attack is. If I understand the term "injection ratio" well (I did not find a definition), a 10% injection ratio would mean adding fake resumes that amount to 10% of the total resume count. On a platform like LinkedIn, with almost 1 billion users, this would mean 100 million fake resumes and accounts. I do not see how this can work with the attack setting that _"Attackers can only inject a limited number of fake resumes
to evade the detection by the platform’s security mechanism
(e.g., fake resume filtering)"_ (S3.2). Surely, the platform must be able to detect such an enormous influx of fake accounts. Even at the "tiny" injection ratio of 0.1%, this still means 1 million fake resumes are created. I also do not see how this would not completely overwhelm the number of genuine resumes that list a specific company.

I found section 4 rather hard to follow, especially as several symbols were seemingly used without proper definition, e.g., `P` and `L` in 4.4.

In the evaluation (section 5), only relative improvement rates are given. I expect at least some absolute hit ratios to be provided as well, to gauge how effective the attack is in practice (not just that the proposed model is better: 10x something very bad will not become very good).

In terms of novelty, it is not fully clear to me whether there is a difference between the concrete technical part of the attacks proposed and the "proliferation of data poisoning attacks on online casual network platforms" (Section 1). While the attack scenarios may be different, the paper does not really address whether there are also any novel technical components of the attacks used.

In terms of ethics, I would like to know whether users of the job platform agreed to their resume information being used for research purposes. (Given that job trajectories are quite unique, purely stripping PII leaves ample room for deanonymization based on the trajectory.)

Smaller comments:
- I would find it useful if S3.3 clearly identifies and names the type of victim(s) in each attack scenario.
- Table 1 can be fully inline in the text, saving some space for other elaborations.

**I have read the rebuttal.**

**Questions:**

- Explain whether the attack scenario is realistic, given the high absolute number of fake resumes required to achieve the injection ratio.
- Please give some examples of the absolute hit ratios achieved by your approach.

**Reviewer Confidence:**

2: The reviewer is willing to defend the evaluation, but it is likely that the reviewer did not understand parts of the paper

**Scope:**

3: The work is somewhat relevant to the Web and to the track, and is of narrow interest to a sub-community

---

### Official Review · Reviewer_NAiH · 2023-11-22

**Novelty:** 4
**Technical Quality:** 4

**Review:**

Summary:
The paper uncovers vulnerabilities in online job platforms like LinkedIn and Indeed due to data poisoning attacks. The study introduces three attack scenarios manipulating company and user recommendations. The authors develop a "fake resume" generation framework, FRANCIS, demonstrating that data poisoning significantly distorts matchmaking results, posing a potential threat to the integrity of services on job platforms.

Strengths:
+ Available artifacts.
+ Well-written and easy to follow.
+ Disclosed unknown vulnerabilities on online job platforms.

Weaknesses:
- Presentation could be improved.
- Missing some details.
- Generalization limitations.

**Questions:**

Overall, I enjoy reading this paper. The idea is interesting and it is easy to follow with specific attack scenarios (company promotion, company demotion, user promotion). However, I have several concerns as follows.

How do you collect data? Is it easy to crawl data from a career platform? Is that allowed under their terms of use? If not please add a warning for the readers.

Generalization limitations: The paper may not thoroughly discuss the generalizability of its findings across different job platforms or the potential variations in vulnerability levels. A broader discussion on these aspects would strengthen the paper's applicability.

The presentation can be improved.
(1) The RQs in evaluation are clearly claimed, however, there is no clear answer for each RQ. Better to make a summary of the results of each RQ at the end of each subsection.
(2) Why are 3 formulas labeled with a serial number after them and not the others?

**Ethics Review Description:**

Ethical aspects could be mentioned. For example, is crawling data allowed by the career platforms?

**Ethics Review Flag:**

Yes

**Reviewer Confidence:**

2: The reviewer is willing to defend the evaluation, but it is likely that the reviewer did not understand parts of the paper

**Scope:**

4: The work is relevant to the Web and to the track, and is of broad interest to the community

---

### Official Review · Reviewer_4EVo · 2023-11-24

**Novelty:** 3
**Technical Quality:** 3

**Review:**

I thank the authors for their submission to WebConf. I found the paper to be well-written and easy to read. In this paper, the authors propose 3 new attacks against resume websites: artificially promoting a target company to its users, artificially demoting a target company to its users, artificially promoting a target person to a company. The experiments using previously proposed job mobility prediction platforms (such as NEMO and AHEAD) were well constructed. I also liked the comparison with multiple baseline models. The one issue I saw regarding baseline models was with the proposed GPT-4 baseline model. Is it possible that the prompt can be improved by adding more details and providing few-shot examples? The currently provided prompt seemed a little rough. For example, the statement “our primary objective is to increase the likelihood of following target companies by adding them to HR models” seems very confusing even for a human reader given that term “HR” models was not even explained before.

But the fatal issue for me regarding this paper is the likely infeasibility of the proposed attacks in real-world scenarios. Tables-3, 4 and 5 detail the results of the proposed attack against NEMO [16]. Taking “user promotion attack” as an example, the authors’ proposed attack model increases the hit rate by 11% (IR = 1.11), if 0.1% fake profiles get added. On a website such as LinkedIn which has close to a billion users, this requires about 1 million fake profiles to be created on LinkedIn. This seems like too much of an effort for any attacker for too little gain (a 10% increase in their visibility). Furthermore, LinkedIn’s bot detection strategies might simply make creation of such a number of accounts impossible (based on IP address, the absence of realistic user activity in this 1-million accounts etc.) [A]. Note that this injection ratio is in fact is the lowest considered in the paper while the highest injection ratio of 10% requires creation of 100 million user accounts which simply appears to be too expensive for a single attacker.  The real-world example of fake SpaceX engineers cited in the footnote of the paper only mentioned 1000 fake accounts which indicates an injection ratio of 0.0001% almost two orders smaller than the smallest injection ratio considered in the paper.

[A] https://about.linkedin.com/transparency/community-report#fake-accounts

**Update after rebuttal**
My concerns remain after discussion with the authors. While I understand that "abuse-prevention" methodologies are outside the scope of this work, my reading is that the fact that they exist ([A]) makes the proposed attacks very non-realistic. Abuse-prevention methodologies such as CAPTCHAs are "low-cost" technologies that most websites deploy. For example, I tested creating an account on the "mid-tier" bizreach.jp domain that the authors mentioned in their discussion. It asked me to give an e-mail address to which it sent a verification code. So, that means, for an attacker to create 1000 accounts secretly on bizreach, they will need to acquire 1000 e-mail accounts first. Which means they will need to create 1000 accounts on Google or Yahoo first which requires beating their abuse-prevention methodologies as well! (Creating 1000 emails accounts on an attacker-owned domain would cause undue suspicion to the target website.) The authors need to keep these practical issues that attackers face in mind when considering the reality of their proposed attacks. As it stands now, unfortunately, it is not convincing that these attacks present a realistic threat to even mid-tier career websites.

**Questions:**

1. Are the considered injection ratios realistic? Please see above for details.

**Reviewer Confidence:**

2: The reviewer is willing to defend the evaluation, but it is likely that the reviewer did not understand parts of the paper

**Scope:**

4: The work is relevant to the Web and to the track, and is of broad interest to the community

---

### Official Review · Reviewer_ZDrf · 2023-11-25

**Novelty:** 5
**Technical Quality:** 4

**Review:**

The authors developed and tested a system to create fake resumes that can be used in three types of data poisoning attacks: company promotion, company demotion, and user promotion. The paper did not clarify why a user demotion attack is not considered, where, for example, an attacker is targeting a set of users, e.g., users with job experiences consistent in certain countries.

The authors didn't discuss the pros and cons for users to engage in a user promotion attack - which is risky, takes a lot of effort, and potentially, candidates would still be filtered at a later stage of the review process – compared to simply creating a fake resume.

The authors made multiple design choices that were not well explained. The authors randomly selected resumes with at least five legitimate work experiences. Why 5 and not 3, 4, or 6 was used as the selection threshold? Why did the study focus only on the tech and business sectors? Why were companies filtered out that only appeared once?

The authors show that their framework FRANCIS outperforms baseline models significantly in nearly all tests. Next to the main metric (improvement rate) used, it would have benefited the paper if the authors included the average hit rate. A reader might wonder how much an attacker gains for running a potentially complicated and expensive data poisoning attack on a large platform.

The paper considered a relatively small dataset of about ten thousand resumes and companies. However, large platforms have many more users (e.g., LinkedIn has hundreds of millions of active users, and Indeed reports numbers in the same vicinity). On these platforms, models are trained on vastly larger datasets; thus, poisoning requires more fake profiles. For example, a 0.1% injection rate on LinkedIn would mean 300,000 fake profiles are required.
Given these platforms' models and email and phone verification requirements for account creation, creating tens of thousands of fake profiles could be expensive.

An additional consideration is whether a successful attack is worth it. Assume a company appears as a recommendation for more users; thus, more users apply. However, the original model better predicts which jobs these users are likely to take. As a result, these additional applicants might eventually not be a good fit for the company, creating more work for HR. Further, users might use the data poisoning company to have competitive offers but eventually choose a career path as predicted by the original model, also creating more work for interviewers.

**Questions:**

Why did you not consider user demotion attacks?

What are the risks and costs vs the benefits of engaging in these data poisoning attacks for different entities in the ecosystem?

Why were five job experiences used, and why weren't 3, 4, or 6 used as the selection threshold? Why did the study focus only on the tech and business sectors? Why were companies filtered out that only appeared once?

What was the average hit rate for the different models?

Would your attack work on a model trained on big data on large platforms? What would be the cost of such an attack? Would it be feasible and profitable?

How could a job platform best defend against such data poisoning attacks?

**Reviewer Confidence:**

3: The reviewer is confident but not certain that the evaluation is correct

**Scope:**

4: The work is relevant to the Web and to the track, and is of broad interest to the community

---

### Decision · Program_Chairs · 2024-01-22

**Decision:**

Accept

**Comment:**

The paper proposes an interesting attack to a vertical sector: Online job platforms. In particular, the authors show that the attack is feasible, and has impact. However, the scale of the evaluation seems too small to provide representative results.
 The reviewers highlighted pros and cons, and the discussion with the authors was quite vivid. When reviewers are assessed also as a function of their confidence, the paper ranks higher.
 IMHO, I believe that the shown attacks are interesting and novel (for the application field; data poisoning is not novel in itself), though the technical quality is not particularly high.

 ---